# Emerging Role of YAP and the Hippo Pathway in Prostate Cancer

**DOI:** 10.3390/biomedicines10112834

**Published:** 2022-11-07

**Authors:** Filippos Koinis, Evangelia Chantzara, Michael Samarinas, Anastasia Xagara, Zisis Kratiras, Vasiliki Leontopoulou, Athanasios Kotsakis

**Affiliations:** 1Department of Medical Oncology, University General Hospital of Larissa, 41221 Larissa, Greece; 2Laboratory of Oncology, Faculty of Medicine, School of Health Sciences, University of Thessaly, 41500 Larissa, Greece; 3Department of Urology, General Hospital “Koutlibanio”, 41221 Larissa, Greece; 43rd Urology Department University of Athens, “Attikon” University General Hospital, 12462 Chaidari, Greece

**Keywords:** prostate cancer, YAP/TAZ signaling, Hippo pathway, prostate cancer hallmarks, YAP

## Abstract

The Hippo pathway regulates and contributes to several hallmarks of prostate cancer (PCa). Although the elucidation of YAP function in PCa is in its infancy, emerging studies have shed light on the role of aberrant Hippo pathway signaling in PCa development and progression. YAP overexpression and nuclear localization has been linked to poor prognosis and resistance to treatment, highlighting a therapeutic potential that may suggest innovative strategies to treat cancer. This review aimed to summarize available data on the biological function of the dysregulated Hippo pathway in PCa and identify knowledge gaps that need to be addressed for optimizing the development of YAP-targeted treatment strategies in patients likely to benefit.

## 1. Introduction

The life-prolonging, widely used, and approved therapies for prostate cancer (PCa) are based on targeting androgen signaling, bone targeting with radiopharmaceuticals, and chemotherapy. Nonetheless, the clinically relevant targets in PCa management have not changed over the years, and current treatment options have already reached their potential. Refinement in existing strategies will likely result in incremental improvement in survival rates, but major advances in therapy will likely require novel treatment approaches founded on an alternative biological basis [1,2]. In line with this are the encouraging observations reported with immune checkpoint blockade in select subsets [3], as well as with the use of PARP inhibitors in cancers with DNA damage repair alterations [4,5]. However, despite the marked lengthening of overall survival (OS), cure rates in men with metastatic PCa remain low [6]. The long-term goal of therapy is to shift from the prevailing treatment paradigm to a curative paradigm. To achieve this transition the understanding of alternative mechanism(s) that result in the emergence of resistance to therapy will be required.

A growing amount of research suggests that the Hippo pathway is part of a network of interlinked signaling pathways that regulate cell transformation, proliferation, invasion, migration, metastasis, and ultimately cancer progression. Increasing our understanding of and identifying the relevant nodes in the dynamic molecular cascade of the Hippo pathway might unveil tumor vulnerabilities that could be therapeutically exploited. Sufficient observations exist to suggest that the key mediator of the Hippo pathway, Yes-associated protein (YAP), is implicated in the emergence of resistance to treatment in PCa. In this review article, we summarize the emerging evidence linking YAP to PCa development and resistance to treatment.

### 1.1. YAP Protein and Hippo TAZ–YAP Signaling Pathway under Physiological Conditions

YAP was initially identified in the late 1990s [7] in Drosophila melanogaster by screening for loss-of-function mutations resulting in organomegaly, due to increased cell proliferation. However, research interest on elucidating its role remained poor until 2005, when it was realized that it was an ortholog of Drosophila Yorkie, the nuclear mediator of the Drosophila Hippo pathway [8]. The Hippo signaling cascade is an evolutionarily highly conserved pathway in mammals regulating organ growth by limiting cell proliferation under high cellular density environmental conditions [9,10,11,12,13]. Recent advances in genome and proteomic sequencing technologies combined with novel bioinformatic tools have successfully uncovered the key components of the Hippo pathway and prioritized research projects have focused on delineating the cross-links with other molecular pathways.

YAP is a 488 amino acid phosphoprotein that acts as a transcriptional co-regulator. Structurally, YAP protein and its paralogue protein TAZ contain a proline-rich, WW domain, which consists of two conserved tryptophan residues separated by 20–23 amino acids [14]. The WW domain of YAP/ΤAΖ recognizes and binds to PPXY motif (proline/proline/any amino acid/tyrosine) found in a variety of proteins, known to control YAP/TAZ localization and function. Although YAP and TAZ share similar amino-acid sequences (~60%), the latter contains one WW domain while YAP contains two. This feature, among others, underlines the fact that there are also fundamental differences between the two molecules [15], reflecting distinct functional properties [16], as well. Moreover, YAP contains a TEAD-binding domain [17] and a C-terminal PDZ-binding motif [18,19], which binds to zonula occludens 1/2 and Na+ /H+ exchange regulatory cofactor proteins and is necessary for its nuclear localization [20,21,22,23]. As a transcriptional co-regulator having both co-activator and co-repressor capabilities, YAP is one of the nuclear-effector factors of the Hippo signaling pathway [24,25].

The Hippo pathway in Drosophila consists of a protein network, including two serine/threonine kinases Hippo (Hpo) [26,27,28] and Warts (Wts) [29], the adaptor proteins Salvador (Sav) and Mob (Mats) [30,31], a transcriptional co-activator Yorkie (Yki), and the transcriptional factor Scalloped [32,33]. Yki is inhibited by the Hippo cascade and is considered the principal functional component of the Hippo pathway [8]. Remarkably, multiple mammalian homologs correspond to a single drosophila protein, pointing out the great complexity of the human Hippo pathway [34].

The core kinase cascade of mammalian Hippo pathway comprises of upstream Ste20-like kinases MST1/2 (Hpo in drosophila), scaffold protein SAV1 (Salvador homolog), large tumor suppressor kinases LATS1/2 (Wts homologues), and protein kinases MOBKL1A and MOBKL1B (Mats homologs), also referred to as MOB1. These proteins create a kinase cascade leading to nuclear localization of YAP and TAZ (Yki homologs). The Hippo kinase cascade is triggered either by TAO kinases TAOK1/2/3 [35], which phosphorylate MST1/2 and cause its activation or by MST1/2 autophosphorylation [36,37]. Both SAV1 and MOB1 function as co-activators of MST1/2 and LATS1/2, respectively. SAV1-bound MST1/2 phosphorylate and activate MOB1-bound LATS1/2 kinases which in turn phosphorylate YAP, promoting cytoplasmic localization and inhibiting nuclear translocation [37,38,39,40]. Coincidentally, other kinases, such as MAP4K, WNT, PI3K, and G-protein-coupled-receptor family kinases have been reported to phosphorylate and activate LATS1/2 [41,42,43,44,45,46]. Activated LATS1/2 phosphorylate YAP at five sites (S61, S109, S127, S164, S381) and TAZ at four sites (S66, S89, S117, S311) [47,48]. Phosphorylated YAP (on the serine 127 site) are secluded in the cytoplasm by binding to 14–3–3 protein and degraded by the ubiquitin–proteasome pathway, leading to inhibition of the downstream transcriptional programs [34,49,50]. Therefore, the Hippo pathway represents a dynamic signaling network, with YAP/TAZ under constant phosphorylation and dephosphorylation. Indeed, there is evidence that YAP is shuttling between nucleus and cytoplasm and may be partially nuclear or partially cytoplasmic [51,52]. Moreover, as YAP/TAZ are co-regulators and do not comprise DNA-binding domains, they require a binding partner protein to regulate gene transcription. The primary transcriptional partners of YAP/TAZ are the TEAD family of transcription factors (Scalloped ortholog), but YAP also associates with other DNA-binding proteins, such as SMAD family transcription factors, RUNX1/2, ErbB4, TBX5, and p63/p73 [53,54,55]. For example, YAP/TAZ bind to TEAD, inducing expression of a variety of target genes implicated in cell proliferation and migration (e.g., *AREG*, *CTGF*, *Cyr61*, *FGF1*, *AXL*, *BMP4*, *PD-L1*) [56]. When YAP/TAZ are absent, VGLL4 protein (transcription cofactor vestigial-like protein 4) binds to TEAD and restrains target gene expression [57]. Thus, when the Hippo pathway is “switched on”, phosphorylation of YAP/TAZ results in inhibition of tissue growth and cell proliferation. On the contrary, when the pathway is “switched off”, YAP/TAZ are dephosphorylated and translocate into the nucleus (Figure 1).

Concluding, the Hippo signaling pathway governs activity of the transcriptional co-activators, YAP/TAZ, and thus expression of genes involved in cell proliferation and migration. Therefore, under physiological conditions, the Hippo pathway regulates tissue homeostasis, organ size, and regeneration.

### 1.2. YAP Protein and Hippo TAZ–YAP Signaling Pathway in Cancer

Given the fact that, under canonical conditions, YAP is involved in cell proliferation, stem cell regulation, and organ development and regeneration, it is expected that Hippo pathway is implicated in tumorigenesis. Indeed, immunohistochemical analysis of a wide variety of tumor samples revealed that YAP overexpression is a common feature among multiple neoplasms, including colon, pancreatic, hepatocellular, gastric/esophageal, ovarian, brain, breast and lung cancer [58]. Moreover, expression of YAP has been linked with poor clinical outcomes. A meta-analysis by San et al. revealed that YAP overexpression is associated with worse disease-free survival (DFS) and overall survival (OS) in large datasets of patients with different types of solid tumors, including ovarian, endometrial, breast, colon and hepatocellular carcinomas [59]. Similarly, recent reports have demonstrated the association of YAP/TAZ overexpression and poor prognosis in non-small cell lung cancer (NSCLC) [60,61], colorectal [62,63], gastric [64,65], esophageal [66], breast [67], liver [68], urothelial [69], and endometrial cancer [70,71]. Notably, as nuclear localization of YAP serves as a surrogate for activated Hippo pathway, high nuclear YAP expression correlated better with shorter survival [59]. Further establishing the functional relevance of elevated YAP levels in cancer, nuclear YAP is more common in neoplastic tissues than cytoplasmic YAP, which is found in 85% of normal tissues [72,73,74]. In line with these observations, a growing amount of evidence supports that YAP can exert both pro-tumorigenic and tumor suppressor effects depending on the cellular localization and YAP’s interacting partners [75].

Although the Hippo pathway is frequently dysregulated in a wide variety of human malignancies, mutations in YAP or other nodal components were considered rare, except for NF2. Loss of function NF2 mutations are linked with the development of type 2 neurofibromatosis, a disorder characterized by increased risk of developing malignant and benign tumors in the nervous system [76]. However, as genome sequencing techniques are evolving, detection accuracy and sensitivity have both improved and more genetic aberrations are being identified. Under this prism, the recent evaluation of more than 9000 samples from different tumor types revealed a relatively high rate of YAP/TAZ gene amplification in squamous cell carcinomas, mostly head and neck and gynecologic tumors. Furthermore, mutations in YAP or TAZ were rare, but functionally important. Interestingly, YAP1/TAZ gene fusions have also been described in several rare malignancies, including porocarcinomas, epithelioid hemangioendotheliomas, ependymomas and meningiomas. These fusion proteins include YAP1-TFE3, TAZ/CAMTA1, YAP1-MAMLD1, YAP1-MAML2, and YAP1-NUTM1 gene fusion [77,78,79]. These fusion proteins have been shown to permanently localize in the nucleus and thus avoid degradation, leading to excessive activation of the Hippo pathway signaling [80], which is unresponsive to the LATS-mediated negative feedback loop.

Since genetic abnormalities are probably responsible for a small portion of increased YAP expression and activity in human cancer, the Hippo pathway’s tumorigenic capacity builds upon its ability to orchestrate multiple biological processes through a variety of molecular mechanisms, including direct activation of oncogenic transcriptional factors, crosstalk with other signaling pathways and inducing pro-tumorigenic changes in the tumor microenvironment [81,82]. In fact, it enables the acquisition of certain properties by the malignant cells, most of which are defined as Hallmarks of Cancer [83]. Foremost, its role in promoting cell proliferation via overexpression of various genes involved in cell cycle regulation, DNA repair and replication, is well established [13,84]. However, YAP also contributes to carcinogenesis through enhancement of tumor cell metastatic potential by unlocking cell plasticity, promoting epithelial-to-mesenchymal transition [85] via induction of ZEB1/2 expression and by inducing angiogenesis via upregulation of the VEGF pathway signaling [86]. Furthermore, it mediates resistance to treatment via inhibition of apoptosis through TEAD-mediated stimulation of anti-apoptotic gene expression [32] and reduction of anoikis, a type of cell death that occurs when cells detach from the extracellular matrix and is considered a barrier to metastasis [31,34,87,88]. Inversely, disruption of YAP–TEAD interaction with verteporfin restored apoptotic death and inhibited cell proliferation [89]. Recently, YAP has been reported to coordinate a transcriptional cell reprograming leading to an extensive metabolic rewiring that is crucial for continuously supplying the energy and nutrients needed to maintain a high rate of proliferation [90]. This deregulation of cellular energetics is mediated by increased expression of metabolic enzymes and nutrient transporters and altered mitochondria functionality, to impact fatty acid, glucose and glutamine metabolism [91]. There are ample data supporting the premise that increased YAP activity is linked with the establishment of a tumor promoting chronic inflammatory response [92]. Kim et al. reported that hyperactivation of the Hippo pathway in murine models increased the expression of markers of inflammation and led to hepatocellular carcinoma (HCC) development. Silencing YAP gene expression significantly reduced hepatic inflammation and inhibited HCC formation [93]. Finally, the YAP–TEAD complex promotes the recruitment of immunosuppressive MDSCs in the tumor microenvironment via upregulation of the CXCL5-CXCR2 signaling. Indeed, targeting YAP-1 has been shown to restore immune destruction of tumor cells, further potentiating the effectiveness of immunotherapy [94].

In conclusion, the Hippo pathway, being at the core of several hallmarks of cancer (Figure 2), potentially represents a nodal vulnerability that should be therapeutically targeted.

YAP expression and Hippo TAZ–YAP signaling pathway regulate several hallmarks of cancer by different mechanisms in various cancer types. The pathway is implicated in cell proliferation, apoptosis, vasculogenesis, invasion, and metastasis. It also interacts with growth suppressors, nucleosome remodeling, and histone deacetylase complexes and under certain conditions induces senescence. Mutations in different components of the pathway have been found functionally important for tumorigenesis. Moreover, increasing evidence suggests a cross talk between YAP and immune system cells that may further result to immune system regulation during cancer. Additionally, different inflammatory factors, gene mutations and epigenetic factor reprogramming during cancer have been found to alter the Hippo TAZ–YAP signaling pathway function. In black the hallmark capabilities, in green the enabling characteristics and in magenta the emerging hallmarks of cancer.

### 1.3. Aberrant Activation of YAP/TAZ in PCa

The PCa cell addiction to YAP hyperactivation and resultantly the biological relevance and clinical significance of the Hippo signaling pathway dysfunction has been confirmed by several studies. There is a growing amount of evidence that YAP mediates PCa initiation, progression, metastasis, transition to castration resistant disease state, and resistance to treatment. YAP overexpression and increased nuclear localization is a common feature among PCa tissue specimens across various disease states [34,95]. However, the underlying molecular mechanisms that govern the dysregulation of the Hippo pathway in PCa have not been fully elucidated.

### 1.4. The Role of Hippo Pathway in PCa Initiation

Zhang et al. reported that ectopic YAP expression triggered malignant transformation and increased cell proliferation, in vitro, in immortalized prostate epithelial cells. Mechanistically, they showed that YAP induces PCa formation in an androgen independent manner, via promoting AKT and MEK-ERK pathway signaling [96]. It is hypothesized that YAP overexpression may lead to constitutive cell proliferation, by bypassing contact inhibition, a process that arrests cell growth when they come into contact. Similarly, Sheng et al. studied YAP1 expression in 62 tissue samples from tumor, tumor-adjacent normal tissue, and benign prostatic hyperplasia and found that cancer cells express higher levels of YAP1 than non-neoplastic cells [97]. However, it should be noted that other studies have reported low levels of YAP by IHC in PCa samples. In an early study investigating the relationship between YAP and PCa, Hu et al. analyzed YAP staining in 66 tumors and reported decreased YAP expression in tumor cells relative to hyperplastic or normal prostate tissue [98]. Nevertheless, Wang et al. reported that although YAP expression is high in basal cells, luminal cells of the normal prostate tissue do not stain positively for YAP. This observation may account for the contradictory results among the various studies. Moreover, they provided a possible link between YAP and PCa tumorigenesis in a Pten/Smad4 deficient in vivo model via induction of an immunosuppressive tumor microenvironment [99].

Furthermore, there is growing evidence that the Hippo pathway is involved in cell polarity loss. Epithelial cells require cell polarity to retain their columnar structure and functionality. Loss of cell polarity, tissue disorganization and uncontrolled cell proliferation are hallmarks of cancer [100]. The partitioning deficient (PAR) complex, which includes Par3, Par6, and atypical protein kinase C (aPKC), the crumbs complex, and the scribble complex have all been identified as the main evolutionarily conserved polarity complexes [101,102]. Specifically, Par3 loss is implicated in the emergence of high-grade prostatic intraepithelial neoplasia (HGPIN), a well-established pre-malignant lesion, by promoting prostatic epithelial cell growth, symmetrical basal cell division and randomizing spindle orientation in luminal cells. In the same study, using knockout mouse models they showed that Par3 loss can inhibit the Hippo pathway via dissociation of Par3/merlin/Lats1 complex. Interestingly, whereas Par3 or Lats1 loss alone can inactivate the Hippo pathway, deletion of either gene can only cause a high-grade PIN but not an invasive–malignant phenotype. Collectively, a combination of Par3 loss and Hippo pathway blockade by co-deletion of Par3 and Lats1 can promote development of PCa [103].

ETS-regulated gene (ERG) overexpression is another mechanism that has been proposed to induce malignant transformation of normal prostate epithelial cells. ERG is a member of the E-26 transformation-specific (ETS) family of transcription factors with diverse functions, such as regulation of cell proliferation, differentiation, vasculogenesis and apoptosis [104]. Abberant ERG overexpression is driven by a translocation and specifically the gene fusion on chromosome 21q22 between the 5′ untranslated region of the androgen-regulated gene TMPRSS2 (acting as a promoter) and the coding sequence of ERG [105]. ETS translocation variant 1 (ETV1) induces YAP activation through a cooperation with lysine specific demethylase (JMJD2A). ETV1 facilitates the recruitment of JMJD2A to the YAP1 promoter, altering the histone lysine methylation in prostate cancer cells. Overexpression of JMJD2A leads to the formation of PIN in mice, which can progress to malignancy [106]. More specifically, ERG seems to promote YAP and TEAD transcription activity and thus induce YAP-target gene protein expression. Immunoprecipitation studies in human luminal-type PCa cells have shown that ERG binds to the promoter of YAP gene, eventually leading to YAP protein expression. In a fundamental study by Nguyen et al., ERG-mediated YAP overexpression in the prostate of healthy mice led to the development of age-related PCA [107], thus linking Hippo pathway dysregulation to PCa initiation.

### 1.5. The Role of Hippo Pathway in PCa Progression and Metastasis

YAP activates the transcriptional activity of various target genes implicated in proliferation, migration, and invasiveness, thus enabling PCa progression through multiple molecular mechanisms. In addition, higher expression of YAP is associated with poor prognosis and shorter patient survival [95]. To this end, several studies have demonstrated YAP’s effect on the acquisition of an aggressive PCa cell phenotype via various in vitro experiments, including cell proliferation, migration, and anchorage-independent growth assays. Zhao et al. silenced YAP and TAZ expression using RNA interference protocols in PCa cell lines PC-3 and DU-145 that have increased metastatic potential. They reported that YAP/TAZ knockdown inhibited both proliferation and cell migration/invasion suggesting the Hippo pathway as an important regulator of PCa progression [87]. Zhang et al. indicated that YAP expression is a potent contributor to the emergence of PCa metastasis. More specifically, they showed that YAP overexpression promotes cell motility and invasion in LNCAP cells through an androgen-independent activation of the androgen receptor (AR) signaling [96]. Moreover, mRNA analysis in PCa patients has revealed significant lower LATS1/2 levels in metastatic compared with clinically localized tumor samples. Thus, suppression of LATS1/2 expression may contribute to YAP-mediated induction of metastasis [87]. On the other hand, Liu et al. focused on the role of TAZ. They reported that TAZ can enhance an invasive PCa phenotype by facilitating the interaction between E26 transformation-specific (ETS) transcription factors and the SH3 domain-binding protein 1 (SH3BP1). Both ETS and SH3BP1 are well-established TAZ target genes [108]. Corroborating the abovementioned data, Collak et al. linked YAP overexpression with extraprostatic extension in patient samples. They also found higher YAP gene copy number in metastatic tissue compared with matched samples from the primary tumor [109]. Further supporting these data, Lee et al. demonstrated an increased YAP expression in the lymph nodes compared with the primary tumor in orthotopic murine models of human PCa. They hypothesized that differential biophysical cues in the metastatic TME may augment YAP expression in metastases. YAP, but not TAZ, knockdown decreased cellular motility. Finally, they identified extracellular matrix (ECM) stiffness, shear stress, and the ROCK–LIMK–cofilin signaling axis as major regulators of the YAP-induced migration through interaction with TEAD that results in altered expression of multiple genes involved in chemotaxis, invasion, and adhesion [110].

Indeed, numerous studies imply that the Hippo pathway is involved in environmental sensing, a crucial process during the metastatic cascade. Interestingly, YAP-driven altered cancer cell gene expression governs the interaction between the mechanical forces of the TME and the migrating PCa cells at the metastatic niche. Cells from bone metastases proliferate and migrate more readily on high-stiffness substrates by inducing YAP/TAZ nuclear localization, whereas cells derived from lymph nodes proliferate and migrate more readily on low-stiffness substrates by forming clusters with high expression of CD44. Thus, YAP may dictate not only the metastatic potential of PCa cells, but the site of metastasis as well [111]. To this end, Tenascin C, a protein secreted by endothelial cells that have undergone endothelial-to-osteoblast transition, increases the metastatic capability of PCa cells in a YAP-dependent manner [112]. Recent data have pinpointed ECM as an important regulator of epithelial cell polarity and provided evidence that dislocalization of polarity proteins, such as dysregulation of Par3, is linked to metastasis [113,114,115,116]. Increased Par3 expression promotes PCa metastasis by inactivating the Hippo pathway through the formation of a noncanonical Par3/aPKC/KIBRA complex. Par3 causes the detachment of kidney and brain-expressed protein (KIBRA) from its typical complex (KIBRA/NF2/FERM domain-containing protein 6 (FRDM6)) and sequentially KIBRA forms a complex with atypical protein kinase C (aPKC). As a result, KIBRA cannot interact with LATS1, leading to LATS1 dephosphorylation and therefore YAP activation [117]. Finally, PCa cells have been reported to secrete a variety of extracellular chaperones, named according to their molecular weight, with HSP90, HSP70, and HSP27 being the most widely studied. HSP27 stimulates ubiquitin-mediated degradation of MST1 that results in reduced phosphorylation of LATS1 and MOB1. As a consequence, YAP is not phosphorylated and its nuclear localization drives PCa progression and increases aggressiveness [118].

### 1.6. The Role of Hippo Pathway in Castration-Resistant Growth in PCa

Androgen-deprivation treatment (ADT) has been proven effective for early-stage hormone sensitive disease, but PCa eventually progresses to an androgen-independent state, leading to bone and soft tissue metastases. Despite recent therapeutic advances, metastatic castration-resistant PCa (mCRPC) remains incurable. This is not surprising given the absence of biomarkers to guide the selection of patients and monitor treatment efficacy or develop more effective therapeutic strategies. Thus, an integral component of efforts mounted to improve clinical outcome of patients with PCa is the understanding of the mechanisms usurped by the cancer to escape addiction to androgens. To this end, several lines of research are being investigated to better understand the biological processes implicated in the emergence of resistance to ADT and progression to CRPC. The main premise being considered is that sustained androgen receptor (AR) signaling remains the main driver of CRPC. Nonetheless, at a cellular level, complex interactions between several signaling pathways governed by central molecular nodes as well as the role of the TME have also been considered to function towards the establishment of CRPC and maintain pro-tumorigenic, both AR-dependent and AR-independent, signaling. Among the multiple mechanisms that have been described, the Hippo pathway seems to represent a key regulator of the expression of many transcriptional factors in the setting of CRPC. Thus, YAP has been traced at the roots of CRPC pathogenesis and drug resistance.

Interestingly, the Hippo pathway has been shown to interact with the AR and therefore modulate its activity regardless of the presence of androgens in the TME. Co-immunoprecipitation studies have confirmed that YAP co-localizes with AR in the nucleus, acting as a co-activator that regulates AR-target gene expression, both in androgen-dependent and androgen-independent PCa cell lines. In particular, MST1 signaling was suggested to directly inactivate and prevent nuclear localization of YAP, independently from LATS status. Attenuation of MST1 activity led to YAP activation, which in turn induced the expression of AR-targeted genes, despite androgen deprivation and treatment with AR-inhibitor enzalutamide. Thus, YAP can promote castration-resistant PCa progression [119]. Indeed, RT-qPCR and western blot analysis have revealed high YAP expression in an AR-null, mimicking CRPC, PC3 cell line [97]. Further confirming these observations, our group showed relatively increased YAP protein expression in AR-independent compared to AR-dependent cell lines [120]. Despite the obvious limitations of in vitro studies, Zhang et al. suggested that YAP overexpression per se might be sufficient to transform PCa cells from an androgen-sensitive to a castration-resistant state. Moreover, they demonstrated that, during this transition, YAP expression was transcriptionally increased, which in turn increased the expression of the AR-targets PSA, NKX3.1, PGC-1, and KLK2, implying that YAP increases AR transcriptional activity in an androgen-depleted environment [96]. Notably, MYB proto-oncogene like 2 (MYBL2) overexpression was found to induce YAP expression and nuclear localization via RhoA activation mediated by Rac GTPase-protein 1 (RACGAP1). This led to the emergence of resistance to ADT in androgen-sensitive LNCaP cells. Blocking this upstream MYBL2 signaling, either by sh-RNA or treatment with an RhoA or YAP inhibitor, reversed the CRPC phenotype and decreased tumor growth in a castrated mouse model [121]. Furthermore, YAP induces the expression of downstream transcription factors SOX2 and Nanog, well-established core stem cell pluripotency regulators. Jiang et al. demonstrated that activation of these pathways promotes the de-differentiation of PCa cells to stem/progenitor-like cells (PCSC), thereby contributing to the development of CRPC [122]. Similarly, cellular myelocytomatosis (c-Myc) overexpression can bypass androgen dependency and stimulate PCa growth in castrated conditions. As c-Myc is another YAP-regulated gene, YAP activation results in up-regulation of c-MYC. C-MYC regulates, among others, the expression of enhancer of zeste homolog 2 (EZH2), an enzymatic component of Polycomb repressor complex 2 (PRC2), an important regulator of cell growth, survival, and differentiation. EZH2 has been reported to act towards progression to CRPC by inducing expression of genes (ERG, DAB2IP and E-cadherin) linked with androgen independence [123,124]. Interestingly, Xu et al. reported that EZH2 participates as a co-activator in AR-associated complexes to support CRPC growth both in vitro and in mouse models [125], supporting constitutive activation of AR signaling, even in the absence of androgens.

Apart from resistance to ADT and YAP-mediated emergence of CRPC, several lines of research have linked YAP overexpression to resistance to therapy in PCa. Matsuda et al. studied YAP expression in a tissue microarray with 203 cores from 70 patients with high-risk localized PCa that underwent radical prostatectomy after receiving neoadjuvant treatment with complete androgen blockade plus docetaxel. Immunohistochemistry (IHC) analysis revealed strong nucleus-localized YAP staining in resistant tumors, with higher YAP protein expression compared with treatment-naïve or treatment-responsive tumors, suggesting that the Hippo pathway mediates resistance to chemotherapy [126]. More recently, increased YAP activity was detected, in vitro, in enzalutamide-resistant (Enza-R) cells. Indeed, YAP protein expression and transcriptional activity, accessed as expression of downstream target genes, were significantly higher in the Enza-R cells compared with LNCaP parental cells. Further confirmation with in vivo experiments showed that YAP and its transcriptional partner (COUP-TFII) were identified in the extracellular vesicles (EVs) isolated from sera of patients with disease progression after treatment with enzalutamide [127]. Finally, our group reported that a YAP/TBX5 mediated increase in FGFR-FGF pathway activity is implicated in resistance to treatment with cabozantinib in both hormone-sensitive and castration-resistant disease states. IHC evaluation of trans-iliac bone marrow biopsies, obtained at baseline and after six weeks under cabozantinib treatment from patients with PCa, revealed relative increases in expression levels of pFGFR1, YAP, and TBX5, therefore supporting the activation of the Hippo pathway as part of the molecular mechanism of acquired resistance to VEGF/MET-targeted treatment [120].

### 1.7. Targeting YAP in PCa

Growing interest in targeting the Hippo pathway is fueled by the above-described data that demonstrate its fundamental role in promoting PCa initiation, disease progression, stem cell function, metastatic potential, development of CRPC, and resistance to treatment. Nevertheless, to date there is a distinct lack of drugs that effectively target YAP with a proven clinical benefit for cancer patients (Table 1). Due to the intrinsic characteristics and complexity of signaling, targeting YAP has been quite challenging.

In fact, for the time being, direct YAP inhibition is considered impossible. This is no surprise, taking into account the recently elucidated YAP protein structure indicating that YAP is small with a considerable shallow surface, containing one WW and one TEAD-binding domain [128]. Therefore, few regions within YAP might be susceptible to therapeutic targeting and development of small molecules that can bind to YAP is burdensome. Thus, YAP has been deemed “undruggable”. An alternative strategy would be to decrease YAP mRNA levels via gene therapy, as recently tested for treating ischemic cardiomyopathy [129]. However, sufficient observations exist to suggest that total inhibition of YAP activity might not be beneficial, as it might result in increased tumor growth through upregulation of WNT signaling [130]. In line with this are reports showing that a 50% decrease in YAP by heterozygous deletion of YAP is sufficient to prevent tumor formation in genetically engineered mouse models [131]. Although such an approach would the limit potential adverse effects, it should be noted that, thus far, this strategy has not been tested in humans and is considered experimental.

Beyond direct YAP inhibition, many research efforts have focused on indirect targeting through impairing YAP stability and nuclear localization, targeting upstream Hippo signaling, or disrupting YAP-transcriptional factor interaction.

The oncogenic effect of YAP is mediated via its dephosphorylated form, while phosphorylated YAP (p-YAP) interacts with 14–3–3, promoting cytoplasmic YAP sequestration. Thus, several drugs can attenuate YAP activity by inducing YAP phosphorylation and preventing YAP nuclear translocation. Dasatinib and pazopanib can effectively inhibit YAP’s nuclear translocation by increasing its proteasomal degradation [132]. A recent study demonstrated that stimulation of muscarinic receptors 1 and 3 results in YAP overexpression through FAK pathway activation. Therefore, the use of FAK inhibitors increased p-YAP levels and decreased YAP activity and PCa cell growth in vitro [133]. Testing this hypothesis in vivo, multiple trials are currently ongoing examining the use of FAK inhibitors either as monotherapy or combined with docetaxel in CRPC. Our group has also shown that XAV-939, a tankyrase (TNKS) inhibitor, suppressed YAP in PC3 cells by increasing its translocation from the nucleus to the cytoplasm. This effect was not mediated through phosphorylation of YAP, as levels of p-YAP remained unchanged during treatment [120]. Following the increased appreciation of Hippo pathway’s role in PCa, several drug repurposing programs aimed to identify alternative strategies to impair YAP function via screening existing drug libraries. Statins, used for treating hypercholesterolemia, inhibit the 3-hydroxy-3-methyl-glutaryl-coenzymeA (HMG–CoA) reductase and thereby prevent HMG–CoA conversion to mevalonate. A product of the mevalonate cascade is geranylgeranyl pyrophosphate, an enzyme that is vital for activation of Rho GTPases. Notably, Rho GTPases increase the phosphorylation rate of YAP and inhibit nuclear accumulation [134]. Interestingly, a growing amount of data from large prospective observational studies link use of statins with a reduced risk of PCa [135]. In line with this, a recent retrospective study that included almost 250,000 Canadian men showed that not only statins but all lipid-lowering drugs are associated with a reduced risk of metastatic PCa and PCa mortality, supporting a possible cholesterol-based mechanism to regulate YAP [136]. Similarly, some evidence implies that individuals taking the AMPK agonist metformin or of the dipeptidyl peptidase-IV (DPP4) inhibitor sitagliptin, both drugs widely used to treat hyperglycemia, have a lower risk of PCa [137]. In vitro studies have confirmed their ability to directly phosphorylate YAP and thus inhibit its transcriptional activity [138]. However, it should be noted that these associations identified with observational studies may not indicate causal inference. Prospective randomized trials should be conducted to provide convincing evidence and validation for the observational results.

An alternative strategy would be to target upstream effectors of the Hippo pathway. HSP27 has been shown to promote the proteasomal degradation of ubiquitinated MST1. As a result, HSP27 overexpression inhibited downstream the phosphorylation of YAP and induced its nuclear localization [118]. Apatorsen (OGX427) is a second-generation phosphorothioate antisense HSP27 inhibitor, which was recently evaluated in a randomized phase II trial in patients with metastatic CRPC. The combination of apatorsen with prednisone demonstrated encouraging results by doubling the proportion of patients experiencing a PSA decline >50% compared with prednisone alone [139]. Similarly promising results have been observed in preclinical models through targeting IKBKE. IKBKE is a non-canonical I-kappa-B kinase, overexpressed in a subset of patients with PCA, that can modulate AR expression via the Hippo pathway. More specifically, IKBE has been reported to phosphorylate and inactivate LATS1/2, leading to YAP activation. On the contrary, IKBKE inhibition results in an increase in LATS1/2 expression, thus promoting YAP cytoplasmic retention and subsequent degradation. In fact, IKBKE inhibitors suppressed tumor growth in mouse xenograft models of CRPC, suggesting that this strategy warrants further clinical evaluation [140]. Drug-discovery programs focused on Hippo-targeting have identified RAF-1 and ALK as MST2 and LATS inhibitors. ALK inhibition increased LATS activity and levels of p-YAP in vitro [141]. Furthermore, alectinib, a novel ALK inhibitor, significantly reduced tumor cell growth in a neuroendocrine ALK F1174C-expressing PCa model [142]. Contrarily, despite the initial enthusiasm generated by the activity of RAF-1 expression inhibitors in vitro, a randomized phase II trial in patients with CRPC reported no benefit [143].

Given that YAP lacks DNA-binding activity and can only act as a co-activator of other transcription factors, inhibition of this interaction represents an alternative approach to indirectly target the Hippo pathway. As TEAD is the main transcriptional partner of YAP, research efforts have been mounted to impair YAP–TEAD complex formation. Moreover, targeting downstream of the YAP–TEAD complex represents an appealing vulnerability within the last steps of the Hippo pathway compared with upstream molecules that are more interconnected with other molecular pathways. Verteporfin (VP) is a small lipophilic benzoporphyrin derivative, used for the photodynamic therapy of macular vascular degeneration. Liu-Chittenden et al. reported that VP selectively binds YAP, changes its structure and disrupts YAP–TEAD interaction [144]. As a result, VP inhibits YAP-mediated PCa cell (PC3) growth and colony formation, without light activation [145]. An actively recruiting phase I clinical trial involving patients with recurrent PCa is currently testing this approach (NCT03067051). Since targeting the YAP–TEAD network represents an intriguing therapeutic opportunity, with minimal off target adverse effects, drug discovery programs focused on developing selective novel agents that can potently inhibit this interaction that is regulated by a protein–protein interaction (PPI). In a pivotal research work, Zhang et al. described the synthesis of a cyclic peptidomimetic small molecule inhibitor of YAP–TEAD interaction and confirmed its therapeutic potential in vivo in a hepatocellular carcinoma xenograft model [146]. Optimization of the research and drug development process led to the introduction of selective small-molecule inhibitors that selectively block TEAD auto-palmitoylation, inhibit YAP–TEAD protein binding and impede NF2-deficient mesothelioma cell proliferation both in vitro and in vivo [147]. Considering that NF2 loss has been linked with PCa progression and poor prognosis [148], patients with metastatic NF2-deficient CRPC may be included in an ongoing basket clinical trial evaluating a novel TEAD inhibitor (NCT04665206). Furthermore, YAP competes with other proteins that can also bind to TEAD. Interestingly, Vestigial-like 4 (VGLL4) is an antagonist of YAP/TEAD activity, which can effectively inhibit YAP-driven TEAD expression and suppress PCa tumor growth when overexpressed [149]. To this end, Jiao et al. managed to develop a VGLL4-mimicking peptide able to interact with TEAD and thus limit YAP activity in cell lines of multiple tumor types, including PCa [150]. Thus far, the efficacy of the above-described strategies has shown promise, but these clinical observations need confirmation and the development of predictive biomarkers that will lead to a truly-targeted application of YAP-targeting therapy.

**Table 1 biomedicines-10-02834-t001:** Proposed strategies to target the Hippo pathway in prostate cancer.

Target	Agent	Method of action	Reference
Tyr	Dasatinib, pazopanib	Increase proteasomal degradation of YAP/TAZ	[132]
FAK	FAK inhibitors	Increase p-YAP levels	[133]
Tankyrase	XAV-939	Increases YAP’s translocation to cytoplasm	[120]
HMG–CoA reductase	Statins	Inhibit nuclear accumulation of YAP	[134,136]
AMPK	Metformin	Inhibits transcriptional activity of YAP	[137,138]
DPP4	Sitagliptin	Inhibits transcriptional activity of YAP	[137,138]
HSP27	Apatorsen	Inhibits nuclear localization of YAP	[139]
IKBKE	IKBKE inhibitors	Increases LATS1/2 expression, promotes cytoplasmic retention of YAP	[140]
ALK	Alectinib	Increases LATS activity and p YAP levels	[142]
YAP	Verteprofin	Disrupts YAP/TEAD interaction	[144]
TEAD	TEAD inhibitor	Inhibit YAP–TEAD protein binding	[147]

## 2. Future Directions

Collectively, various novel and “repurposed” drugs are currently preclinically identified and may soon be tested in randomized clinical trials. However, knowledge gaps and inherent limitations of the Hippo pathway have limited our ability to translate the encouraging preclinical observations into clinically relevant benefits in cancer patients. Chief among these barriers to progress are the perplexing interplay of the Hippo network with other molecular pathways in humans and the inability to create preclinical PCa models that can reflect the complexity of human PCa with fidelity. Moreover, there is no assay to measure biologically relevant YAP activity in a certain patient, and thus there is, to date, no biomarker that will predict for clinical relevance of YAP overexpression in patients with PCa. Improving the understanding of the Hippo pathway and delineating the role of its core kinases, individually and collectively, will be necessary to better appreciate their role, elucidate their contribution to PCa growth and progression, and finally select the most promising therapeutic targets. Further study should also shed light on the mechanism dictating the translocation of YAP to the nucleus providing another target for treatment with possibly fewer side effects. Conclusively, future research on the Hippo pathway should include (1) integration of functional genomics analysis data to fully characterize the Hippo pathway subsets and gain a better understanding of their function in PCa, through both preclinical and translation research projects and validate experimental observations that implicate YAP in PCa progression and resistance to treatment; (2) integration of multi-omics data for the development of biomarker assays in the TME and the peripheral blood of the patients to efficiently monitor Hippo pathway activation; (3) application of these biomarker assays to stratify patients by their “Hippo pathway status” and selection of those for whom YAP is “driving” PCa progression; (4) development of novel strategies to efficiently target YAP-driven PCa progression in small biologically informed proof-of-principle clinical studies that employ biomarker-enrichment strategies and test the clinical relevance of targeting the Hippo pathway while increasing confidence in the chances of a successful phase III trial; (5) international collaborations in an effort to involve larger cohorts and diversified PCa patient populations in launching randomized clinical trials that will evaluate the impact of these strategies on PCa patient survival.

## 3. Conclusions

Converging clinical and experimental observations suggest that YAP is a driving force of PCa initiation and progression, contributing to resistance. The findings account for the interest in the potential impact of persistent YAP expression on treatment efficacy. To monitor YAP clinically, we must understand its role in PCa models that accurately reflect the complexity of human PCa and estimate the utility of a candidate marker of YAP activity in tissue and blood. Furthermore, elucidating the role of key components of the Hippo pathway may lead to the development of specific treatment strategies that exploit the unique vulnerabilities of YAP-mediated resistance to treatment.

## Figures and Tables

**Figure 1 biomedicines-10-02834-f001:**
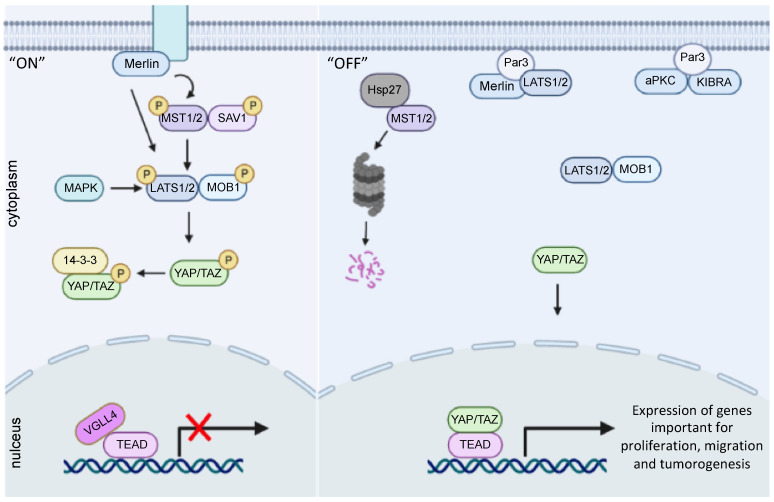
Hippo YAP/TAZ signaling pathway in early PCa. When the Hippo pathway is switched ON, phosphorylation of Mst1 and Lats1 by Merlin and MAPK leads to phosphorylation of YAP/TAZ that forms a complex with 14–3–3 protein in the cytoplasm. VGLL4 binds to TEAD binding promoters thus, blocking the expression of genes important for proliferation, migration and tumorigenesis. In the OFF condition, in early prostate tumor cells, Lats1/2 is dephosphorylated due to various mechanisms such as degradation of Mst1/2 by Hsp27, complex formation of Lats1/2 with Par3 and Merlin and dissociation of Lats1/2 and KIBRA complex by Par3. Due to blocking of physiological kinase cascade of the Hippo pathway, unphosphorylated YAP/TAZ enters the nucleus and act as co-regulator of TEAD transcription factors leading to expression of target genes.

**Figure 2 biomedicines-10-02834-f002:**
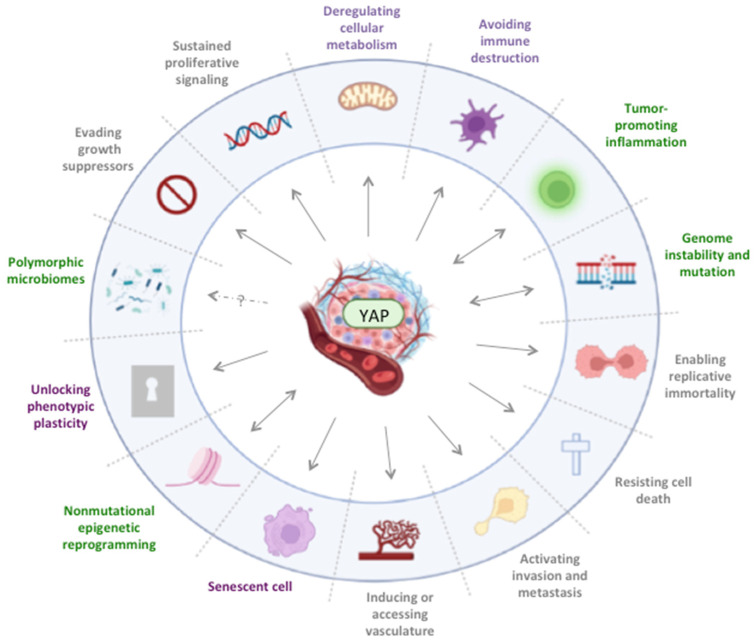
YAP and the Hallmarks of Cancer.

## Data Availability

The data presented in this study are available on request from the corresponding author.

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
