# Peer review of "Emerging Role of YAP and the Hippo Pathway in Prostate Cancer"

_biomedicines, 2022, doi:10.3390/biomedicines10112834_

Round 1

Reviewer 1 Report

In this manuscript, the authors discussed the importance of the Hippo pathway, especially the key player YAP, in prostate cancer development and progression. This review summarized this main point from different angles, including basis mechanisms and treatment strategies. Although this topic has been broadly discussed in the cancer research field, providing new insight into the diagnostic and therapeutical aspects is still critical.

Overall, this manuscript is well structured and written; but a few minor suggestions are below:

1. Adding a model figure that demonstrates the overall molecular mechanisms of the Hippo pathway might help the readers to understand.

2. For the last section, other than the conclusion, more detailed prospectives might be helpful to pave a clear path for future studies.

Reviewer 2 Report

This is generally a well-structured and well-written review that summarises both historical and more recent information on YAP and the HIPPO pathway, specifically with reference to prostate cancer. Selection of articles and information to include in the review is very good as current knowledge and limitations in our understanding of the functioning and potential targeting of the pathway are both described. Although I do not feel as optimistic as the authors with regards to a “pivotal” role of YAP/HIPPO signalling as a therapeutic target in the treatment of prostate cancer, results of experimental studies and clinical trials have been detailed in a balanced manner, thereby allowing readers to critically evaluate the published data and come to their own conclusions. The length of the review is appropriate and I do not feel that inclusion of additional references or information would be essential. For these reasons, I recommend acceptance of the manuscript in its current form.

Minor typographical errors:-

Page 1: “result in emergence”

Page 2: “488 amino acid”  (or “488 aa”)

Page 3: “restrains target gene expression”

Page 3: “Thus, when the Hippo pathway…”

Page 3: “governs activity of the transcriptional co-activators, YAP/TAZ and thus expression of…”

Page 3: “under physiological conditions, the Hippo pathway…”

Page 3: “supports that YAP can…”

Page 3: In sentence starting with “These fusion proteins include….”, the words “gene fusion” at the end of the sentence are redundant and can be removed. (Note also “fusion proteins” (not ‘fusions proteins’)

Page 6: “eventually” (not ‘eveltually’)

Page 6: “fundamental”  (‘fundational” is not English – not sure what intended word is)

Page 8: “Interestingly, the HIPPO pathway has been shown…”

Page 8: “AR-target gene expression”

Page 10: “cholesterol-based mechanism”

Page 10: “downstream of the YAP-TEAD complex”

Page 11: “really-targeted” is not English, but I’m not sure what the intended word is
